# Distribution and Neurochemical Characterization of Dorsal Root Ganglia (DRG) Neurons Containing Phoenixin (PNX) and Supplying the Porcine Urinary Bladder

**DOI:** 10.3390/cells14070516

**Published:** 2025-03-31

**Authors:** Paweł Janikiewicz, Urszula Mazur, Piotr Holak, Nastassia Karakina, Kamil Węglarz, Mariusz Krzysztof Majewski, Agnieszka Bossowska

**Affiliations:** 1Department of Human Physiology and Pathophysiology, School of Medicine, Collegium Medicum, University of Warmia and Mazury in Olsztyn, Warszawska 30, 10-082 Olsztyn, Poland; pawel.janikiewicz@uwm.edu.pl (P.J.); urszula.mazur@uwm.edu.pl (U.M.); kamil.wenglarz@uwm.edu.pl (K.W.); mariusz.majewski@uwm.edu.pl (M.K.M.); 2Department of Surgery and Radiology with Clinic, Faculty of Veterinary Medicine, University of Warmia and Mazury in Olsztyn, Oczapowskiego 13, 10-719 Olsztyn, Poland; 3Department of Anesthesiology and Intensive Care, School of Medicine, Collegium Medicum, University of Warmia and Mazury in Olsztyn, Warszawska 30, 10-082 Olsztyn, Poland

**Keywords:** phoenixin, urinary bladder, sensory innervation, dorsal root ganglia neurons, neuropeptides, immunohistochemistry, pig

## Abstract

The present study was designed to establish the distribution pattern and immunohistochemical characteristics of phoenixin-immunoreactive (PNX-IR) urinary bladder afferent neurons (UB-ANs) of dorsal root ganglia (DRG) in female pigs. The sensory neurons investigated were visualized with a retrograde tracing method using Fast Blue (FB), while their chemical profile(s) were identified using double-labelling immunohistochemistry with antibodies against PNX, calcitonin gene-related peptide (CGRP), calretinin (CRT), galanin (GAL), neuronal nitric oxide synthase (nNOS), pituitary adenylate cyclase-activating polypeptide (PACAP), somatostatin (SOM) and substance P (SP). Nearly half of UB-ANs contained PNX (45%), and the majority of such encoded sensory neurons were small in size (66%). The most numerous subpopulation of FB/PNX-positive neurons were those containing SP (71%). CGRP, GAL or PACAP were observed in a smaller number of PNX-containing UB-ANs (50%, 30% or 25%, respectively), while PNX-positive sensory neurons simultaneously immunostained with nNOS, CRT or SOM constituted a small fraction of all retrogradely-traced DRG neurons (DRGs; 15%, 6.5% or 1.6%, respectively). Furthermore, the numerical analysis of neurons expressing individual antigens, performed on 10 μm-thick consecutive sections, allows us to state that studied sensory neurons can be classified as neurons “coded” either by the simultaneous presence of SP/CGRP/PACAP/GAL, SP/CGRP/PACAP/NOS, SP/CGRP/PACAP/NOS/CRT and/or SP/CGRP/GAL/PACAP, or, as a separate population, those capable of SOM synthesis (SP/CGRP/SOM/PACAP/GAL-positive neurons). The present study reveals the extensive expression of PNX in the DRGs supplying to the urinary bladder, indicating an important regulatory role of this neuropeptide in the control of physiological function(s) of this organ.

## 1. Introduction

Phoenixin (PNX), a bioactive peptide described nearly a decade ago [1], is highly conserved across mammalian and non-mammalian species, with phoenixin-14 and -20 being the most prominent bioactive forms, giving rise to its physiological importance. PNX-14 is identical among multiple species, including human, rat, mouse, dog and domestic pig, whereas PNX-20 differs in one amino acid within the coding region among human, canine or porcine sequences [1].

Initially, this substance was assigned an important function in the regulation of reproductive processes due to its high expression in the rodent hypothalamus and pituitary gland where, acting by one of the orphan receptors, G protein-coupled receptor 173 (GPR173), PNX was able to induce an increase in gonadotropin-releasing hormone (GnRH) receptor mRNA expression and potentiates GnRH receptor upregulation, followed by a subsequent release of luteinizing hormone (LH) or follicle-stimulating hormone (FSH) from cultures of female pituitary cells [1,2]. Further studies have shown that the presence of this peptide is not limited to the hypothalamo–pituitary complex; PNX also contributes to the central control of several other biological processes, including food intake, energy homeostasis, water balance, modulation of inflammation, memory and anxiety (for the corresponding reviews, see [3,4]).

It should be emphasized that PNX is not present only in brain structures. The occurrence of this neuropeptide has also been confirmed in some nervous structures belonging to the peripheral nervous system (PNS). Currently available data, unfortunately concerning almost exclusively the rat PNS, have revealed the presence of PNX-immunoreactive (IR) cells and fibers in, among others, the spinal trigeminal tract and nucleus of the solitary tract, trigeminal and nodose ganglion cells, superficial layers of the dorsal horn (DH) and, last but not least, in afferent neurons of the dorsal root ganglia (DRG) [5]. This latter observation was recently confirmed and further supported by the results of our recent publication analyzing the intraganglionic arrangement and chemical coding of PNX-IR neurons in the DRG of the domestic pig [6]. The observations of Yosten and colleagues, who demonstrated the presence of this peptide in a number of internal organs, such as esophagus, stomach, pancreas, spleen and lung, were an indirect confirmation of the sensory and/or autonomic projections of PNX-positive cells [1].

The sensory component of the innervation of internal organs and its proper functioning is an indispensable part of many processes regulating the activity of human and animal organisms. For instance, DRG sensory neurons of the lumbo-sacral levels of the spinal cord are a crucial part of the afferent part of the micturition reflex arch. Spinal afferent neurons working together with sympathetic and parasympathetic efferent nerves play an important role in reflex control of urine storage and micturition. As reported by Bossowska et al., the sensory information from the porcine bladder is received and transmitted by “dual” afferent pathways, formed by peripheral processes of sensory neurons located, respectively, in the lumbar (L3–L6) and in the sacro-coccygeal (S3–S4 and Cq1) DRGs [7]. Since many urinary bladder disorders have a neurogenic background, it is important to unravel the physiological mechanisms regulating functions of this organ, which, in turn, will allow us to better understand the pathological mechanisms causing dysfunctions of the urinary bladder.

As mentioned above, recent studies regarding PNX have shown that this peptide has widely been distributed in various peripheral tissues. However, to date, there are no data regarding the presence of PNX in the urinary bladder or in the neurons supplying this organ. Therefore, the present study has been designed to investigate (i) the pattern of PNX-IR sensory neurons occurrence in DRG associated with the urinary bladder; (ii) pattern(s) of intraganglionic distribution of PNX-positive neurons in the examined sensory ganglia; and (iii) pattern(s) of colocalization of PNX with other markers of sensory neurons (calretinin—CRT; calcitonin gene-related peptide—CGRP; galanin—GAL; neuronal nitric oxide synthase—nNOS; pituitary adenylate cyclase-activating polypeptide—PACAP; substance P—SP; and somatostatin—SOM) in the bladder sensory neurons of the L as well as the S-Cq neuromeres of the porcine spinal cord. The domestic pig has been chosen for this study, as this species is increasingly employed as an animal model because of the anatomical and physiological similarities to humans with regard to both the structure and function of various tissues and organs [8,9,10].

## 2. Materials and Methods

### 2.1. Laboratory Animals

Investigations were conducted using six immature (8–12 weeks old, 15–20 kg body weight—b.w.) female pigs of the Large White Polish breed. All the animals originated from a commercial fattening farm and were kept under standard laboratory conditions. They were fed standard fodder (Grower Plus, Wipasz, Wadąg, Poland) and had free access to water. The animals were housed and treated in accordance with the rules of the Local Ethics Committee for Animal Experimentation in Olsztyn (affiliated to the National Ethics Commission for Animal Experimentation, Polish Ministry of Science and Higher Education; decision No. 40/2020 from 22 July 2020). All efforts were made to minimize the number of animals used and their suffering. As the present study was designated to provide basic data concerning the chemical phenotypes of the subpopulation of PNX-containing DRG neurons (DRGs) involved in the afferent innervation of the urinary bladder wall under physiological conditions, the authors decided to focus on sexually immature female pigs in order to exclude any possible influences of reproductive hormones on the tissues studied, as identified in previous studies [11,12].

### 2.2. Anesthesia and Surgical Procedures

Before performing any surgical procedure, all the animals were pretreated with atropine (Polfa, Poland, 0.05 mg/kg b.w., subcutaneous (s.c.) injection) and azaperone (Stresnil, Janssen Pharmaceutica, Beerse, Belgium; 0.5 mg/kg b.w., intramuscular (i.m.) injection). Next, buprenorphine (Bupaq, Richter Pharma AG, Wels, Austria, 20 μg/kg b.w.) was given in a single i.m. injection to abolish visceral pain sensation. Thirty minutes later, to induce anesthesia, the main anesthetic drugs sodium pentobarbital (Tiopental, Sandoz, Warszawa, Poland; 0.5 g per animal, administered according to the effect) and ketamine (Bioketan, Vetoquinol, Gorzów Wielkopolski, Poland, 10 mg/kg b.w.) were given intravenously in a slow, fractionated infusion. The depth of anesthesia was monitored by testing the corneal reflex.

A mid-line laparotomy was performed in all the animals (*n* = 6), and the urinary bladder was gently exposed to administer a total volume of 40 µL of 5% aqueous solution of the fluorescent retrograde tracer Fast Blue (FB; Dr K. Illing KG & Co GmbH, Gross Umstadt, Germany) into the right side of the urinary bladder body wall, in multiple injections (1 μL of the dye solution per 1 injection using a Hamilton microsyringe equipped with a 26S gauge needle) under the serosa along the whole extension of the urinary bladder dome, keeping a similar distance between the places of injections. To avoid the leakage of the dye, the needle was left in each place of FB injection for about one minute. The wall of the injected organ was then rinsed with physiological saline and gently wiped with gauze. Three weeks later, the minimum time period needed for the retrograde tracer to be transported to the DRGs and the labeling of urinary bladder afferent neurons (UB-ANs) [7], the animals were deeply anaesthetized (following the same procedure as described above) and, after the cessation of breathing, transcardially perfused with freshly prepared 4% paraformaldehyde in 0.1 M phosphate buffer (pH 7.4). Following the perfusion, all the animals were dissected, and the bilateral DRGs were collected from all the animals studied (as previously described in detail by Bossowska et al. [7]). Tissue samples were then postfixed by immersion in the same fixative (10 min at room temperature), washed several times in 0.1 M phosphate buffer (pH 7.4; 4 °C; twice a day for three days) and finally transferred to and stored in 18% buffered sucrose at 4 °C (two weeks) until sectioning.

### 2.3. Sectioning of the Tissue Samples and Estimation of the Total Number of UB-ANs

Samples of the ipsilateral and the contralateral DRG studied were pairwise located on a pre-cooled block made of a drop of the ‘optimal cutting temperature’ compound medium (OCT) in a manner allowing the easy determination of the position of individual subdomains of DRG sections under the microscope. Frozen ganglia were cut with an HM525 Zeiss freezing microtome on transverse 10 µm-thick serial sections (four sections on each slide), mounted on chrome alum-gelatine-coated slides, air dried and examined under the fluorescent Olympus BX61 microscope equipped with a filter set, allowing the visualization of FB-positive (FB^+^) neurons. To determine the relative number of UB-ANs, FB-positive neurons were counted in every fourth section (to avoid the double-counting of the same neuron; most neurons were approximately 40 μm in diameter) prepared from both the ipsi- and contralateral ganglia of all animals. Only neurons with a clearly visible nucleus were considered. The results were pooled for every experimental animal and statistically analyzed, and the mean number of FB-positive neurons was calculated. The total number of FB^+^ sensory neurons counted in all the ipsilateral and contralateral DRG from a particular animal, as well as the relative frequencies of perikarya in the ganglia belonging to individual neuronal size classes were presented as mean ± standard deviation (SD). The diameter of the FB^+^ perikarya was measured by means of an image analysis software (version 3.02, Soft Imaging System, Münster, Germany), and data were used to divide urinary bladder sensory neurons into three size classes: small (average diameter up to 40 µm), medium-sized (diameter 41–70 µm) and large afferent cells (diameter > 70 µm), while the statistical analysis was performed using Graph-Pad Prism 8 software (GraphPad Software, La Jolla, CA, USA).

### 2.4. Immunohistochemical Procedure

Double-immunofluorescence was performed on cryostat sections of both the ipsilateral and contralateral DRG where the UB-ANs were found, according to a previously described method [7]. The immunohistochemical characteristics of FB^+^ neurons were investigated using primary antibodies raised in different species. After the immersion of the tissues in a blocking solution containing 0.1% bovine serum albumin, 1% Triton X 100, 0.01% sodium azide (NaN_3_), 0.05% thimerosal and 10% normal goat serum in 0.01 M phosphate-buffered saline (PBS) for 1 h at room temperature to reduce non-specific background staining, sections were repeatedly rinsed in PBS, and then incubated overnight at room temperature with PNX antiserum applied in a mixture with antisera against CGRP, CRT, GAL, nNOS, PACAP and SOM (the presence of all the above-mentioned active substances, or their marker enzymes (nNOS), was previously revealed in the porcine UB-ANs [7,13,14,15]. Primary antisera were visualized by fluorescein isothiocyanate (FITC)-conjugated rat- or mouse-immunoglobulin G (IgG)-specific secondary antisera or biotin-conjugated anti-rabbit specific antiserum. The latter complexes were then visualized by application of CY3-conjugated streptavidin. The application of primary antisera raised in different species allowed us to assess the coexistence of the biological active substances investigated. Details concerning all the primary and secondary antibodies used in the present study are listed in Table 1.

Retrogradely labelled and double-immunostained DRG perikarya were evaluated under an Olympus BX61 microscope (Olympus, Hamburg, Germany) equipped with an epifluorescence filter for FB and an appropriate filter set for CY3 or FITC. Relationships between immunohistochemical staining and FB distribution were examined directly by interchanging filters. The images were taken with an Olympus XM10 digital camera (Tokyo, Japan). The microscope was equipped with cellSens Dimension 1.7 Image Processing software (Olympus Soft Imaging Solutions, Münster, Germany).

### 2.5. Estimation of the Chemical Coding of the DRG UB-ANs

All sections containing FB^+^ neurons were used for double-immunohistochemical labelling. To determine the percentages of particular neuronal subpopulations, FB^+^ neuronal profiles were investigated with a combination of two primary antisera on each section (four sections per slide) and counted in both (ipsi- and contralateral) DRG in each animal studied. The relationships between immunohistochemical staining and FB distribution were examined directly by interchanging filters. To avoid the double counting of the same neurons, the retrogradely labeled sensory neurons were counted in every fourth section (only neurons with clearly visible nucleus were included). The colocalization pattern(s) of PNX with other biologically active substances in FB^+^ neurons were examined on consecutive sections. The percentages of FB^+^ neurons that are immunopositive to the individual biologically active substances or their marker enzyme were pooled in all the animals and presented as mean ± SD, with “n” referring to the number of animals. Morphometric data relative to each neuronal class were compared within each animal and among the animals and were analyzed by the Student’s two-tailed t test for unpaired data using GraphPad PRISM 8.0 software (GraphPad Software, La Jolla, CA, USA). The differences were considered to be significant at *p* < 0.05.

### 2.6. Control of Specificity of the Tracer Staining and Immunohistochemical Procedures

The FB injection sites and the tissues adjacent to the urinary bladder were carefully examined macroscopically prior to the collection of DRGs for examination. The injection sites were easily identified by the yellow-labeled deposition left by the tracer within the bladder wall. Moreover, the injection sites were also observed in the UV lamp rays in the dark room. The tissues adjacent to the bladder were not found to be contaminated with the tracer. To verify that the tracer had not migrated into the urethra, we analyzed, in cryostat sections and by means of the H&E staining technique (IHC WORLD, LLC, Woodstock, NY, USA), possible signs of leakage of the tracer to the junction between the urinary bladder trigone and cranial portion of the urethra. No contamination of the urethra with a tracer was found in all the animals studied. All above-mentioned procedures excluded any leakage of the tracer and validated the specificity of the tracing protocol.

To test the specificity of primary antibodies and staining reaction, preincubation tests were performed on sections from the DRG of the control pigs. Preabsorption for the neuropeptide antisera (20 μg of appropriate antigen per 1 mL of the corresponding antibody at working dilution; all antigens were purchased from Peninsula, SWANT, Sigma, Phoenix—Table 2), as well as an omission and replacement of the respective primary antiserum with the corresponding nonimmune serum completely abolished immunofluorescence and eliminated specific labeling.

## 3. Results

### 3.1. Distribution Pattern and Morphometrical Characteristics of FB^+^ Neurons in the Porcine DRG

Three weeks after the administration of FB into the wall of the right side of the urinary bladder, FB-containing sensory neurons were found bilaterally in the DRG of all animals studied. The total number of FB^+^ neurons counted in all DRG per animal ranged from 468 to 593 retrogradely labelled perikarya (527 ± 62.7; mean ± SD). It should be emphasized that a distinct “lateralization” in the localization of UB-ANs was observed in terms of their distribution ipsi- and contralaterally to the sites of FB injections into the bladder wall, with a distinct predominance of FB^+^ sensory neurons in the ipsilateral ganglia. The number of FB^+^ neurons per animal ranged from 428 to 534 retrogradely labelled perikarya (479 ± 53.1) in the DRG of the right side of the body, and from 40 to 59 FB-positive neurons (48 ± 9.8) in the left ones. Approximately 91% of all FB-positive spinal sensory neurons (90.9 ± 0.7%) were located in the ipsilateral ganglia, while only 9% (9.1 ± 0.7%) of retrogradely labelled bladder sensory neurons was observed in contralateral DRGs.

Retrogradely labelled neurons were found to form two distinct “centers”, located bilaterally in the DRG of the L3–L6, S3–S4 and Cq1 neuromeres of the spinal cord. Urinary bladder afferent neurons were, however, virtually absent from the bilateral S1 and S2 DRGs. The vast majority of traced neurons (87.8 ± 1.2% and 86.3 ± 1.9% in the DRG on the right and left side of the body, respectively) were located in S3-S4 and in the Cq1 ganglia, while the remainder of FB^+^ neurons (12.2 ± 1.2% and 13.7 ± 1.9%) were found in the L3–L6 ganglia. Details concerning the relative percentages of retrogradely labelled sensory neurons located in the individual ipsilateral and contralateral DRG are presented in Table 3 and Figure 1.

In terms of the diameter of the UB-ANs and their intraganglionic distribution pattern, they were, in fact, almost identical to those obtained in our previous study (for detailed description, see [15].

### 3.2. Distribution Pattern and Morphometrical Characteristics of FB^+^ and PNX-Containing Sensory Neurons in the Porcine DRG

#### 3.2.1. Distribution Pattern of FB^+^/PNX^+^ Neurons

In general, PNX^+^ UB-ANs were observed bilaterally in all the studied DRG. The total number of FB^+^/PNX^+^ sensory neurons per animal ranged from 214 to 262 (245.3 ± 27.5), representing 45.3 ± 4.6% of all UB-ANs counted in the investigated DRG. The vast majority of FB^+^ and PNX-containing sensory neurons were present in the ipsilateral DRG (82.8 ± 3.3%) while in the contralateral ones, such encoded UB-ANs were observed in much smaller numbers (17.2 ± 3.3%). Furthermore, in both the ipsilateral and contralateral DRGs, PNX^+^ was present in the L (80.9 ± 2.0% and 19.1 ± 2.0%), as well as in the S/Cq (82.9 ± 3.1% and 17.1 ± 3.1%) population of retrogradely labelled DRGs, and there were no significant differences in the number of UB-ANs immunolabelled for PNX between the ipsi- and contralateral DRG studied in these two bladder-supplying “centers”. Nevertheless, it should be noted that the lumbar DRG contained a significantly greater number of FB^+^/PNX^+^ sensory neurons (59.5 ± 4.1%) compared to the DRG forming the S-Cq-supplying “center” (41.2 ± 5.8%).

#### 3.2.2. Morphometrical Characteristics of FB^+^/PNX^+^ Neurons

Regarding the diameter of PNX^+^ UB-ANs, only two classes of these sensory neurons have been determined in all the DRG examined. In both the ipsilateral and contralateral sensory ganglia, a larger subpopulation of small-sized neurons (average diameter up to 40 µm; 65.8 ± 2.0% and 67.1 ± 9.9%, respectively) and the less numerous subset of medium-sized perikarya (diameter 41–70 µm, 34.2 ± 2.0% and 32.9 ± 9.9%, respectively) have been observed, while the immunoreactivity to PNX was absent in large UB-ANs (diameter > 70 µm) of all the DRGs studied. Although in both the L and S-Cq DRGs the majority of retrogradely labelled and simultaneously PNX^+^ neurons belonged to the subset of small-diameter neurons (62.2 ± 8.9% and 71.5 ± 9.3%, respectively), when compared to the number of medium-sized perikarya (37.8 ± 8.9% and 28.5 ± 9.3%, respectively), it should be stressed that in the S-Cq ganglia, small-sized neurons constituted a slightly larger population of PNX-containing DRGs compared to the fraction of medium-diameter neurons.

#### 3.2.3. Intraganglionic Distribution Patterns of FB^+^/PNX^+^ Neurons

To establish the intraganglionic distribution pattern(s) of PNX^+^ UB-ANs in all the DRGs tested, the “mask” shown in Figure 2 was applied to each ganglionic section analyzed.

PNX-containing sensory neurons were present in all five ganglia domains analyzed in this study. In the ipsilateral and the contralateral DRGs examined, the majority of PNX-containing UB-ANs were unevenly distributed, as isolated cells scattered throughout the individual ganglionic subdomains. Only a few retrogradely labelled PNX^+^ DRGs in the ipsilateral ganglia were found to form small, loose clusters up to three neurons.

In the right (ipsilateral) DRG, a distinct accumulation of the retrogradely labelled PNX-positive neurons was found in the caudal (Cd; 36.7 ± 3.6%),in the middle (Md) and the cranial (Cr) ganglionic regions (22.3 ± 1.6% and 20.2 ± 4.2%, respectively), while such labelled sensory neurons were less numerous in the peripheral (P) and central (Cn) domains (12.2 ± 2.1% and 8.6 ± 0.8%, respectively). The distribution pattern of FB^+^ PNX-containing neurons in the DRG contralateral to the site of tracer injections was different to that seen in the ipsilateral ones: the majority of FB^+^/PNX^+^ nerve cell bodies were present in the Cd and P subdomains (26.5 ± 1.5% and 24.8 ± 1.6%, respectively), with the rest of retrogradely labelled PNX^+^ perikarya dispersed quite evenly in the Md, Cn and Cr regions of the ganglion (18.1 ± 4.3%, 17.4 ± 1.4% and 13.2 ± 1.2%, respectively). Details concerning the intraganglionic distribution pattern of retrogradely labelled PNX-positive sensory neurons in the individual domains of the ipsi- and contralateral DRG studied are shown in Table 4.

### 3.3. Immunohistochemical Characteristic of PNX-Containing UB-ANs in DRG Studied

Two main subpopulations of FB^+^/PNX^+^ DRG neurons have been revealed using the technique of double immunohistochemical staining. The vast majority (72.3 ± 2.7%) of all retrogradely labeled, PNX-positive sensory neurons also showed co-occurrence of other substances investigated, although the numbers of cells in individual subpopulations differed from each other. The remaining FB^+^ neurons (27.7 ± 2.7%) were immunopositive only to PNX.

#### 3.3.1. Sensory Neurons Co-Localizing PNX and SP

The largest subpopulation of PNX^+^ DRG bladder sensory neurons were those simultaneously containing SP (Figure 3(A1–A4)). Such labelled neurons accounted for 71.1 ± 5.8% of the total population of PNX-IR UB-ANs found in all examined DRG, while just over a quarter of them (28.9 ± 5.8%) did not concurrently express SP. Statistically significant differences in the numbers of FB^+^/PNX^+^/SP^+^ neurons were neither observed between the ipsi- and contralateral DRGs studied (67.5 ± 5.2% and 76.6 ± 2.6%, respectively), nor between the DRGs forming L and the S-Cq bladder-supplying “centers” (52.2 ± 1.1% and 70.1 ± 3.5%, respectively).

Regarding their diameter, the most numerous subset of simultaneously PNX- and SP-positive UB-ANs was that of small-sized cells (Figure 3(A1–A4); 73.3 ± 4.7%), while such labelled medium-sized neurons were found to be distinctly less numerous (26.7 ± 4.7%). Although the differences in the numbers of both small PNX/SP-IR bladder sensory neurons (70.1 ± 5.2% and 65.3 ± 2.9%) and those of the medium size (29.9 ± 5.2% and 34.7 ± 2.9%) were not statistically significant between the L and the S-Cq DRG, in the ipsilateral ganglia, the statistically significant higher number of the small-sized (85.1 ± 3.9%) and the distinctly lower number of medium-sized (14.9 ± 3.9%) PNX/SP-IR traced neurons were observed when compared with the contralateral ones (70.1 ± 5.2% and 29.9 ± 5.2%, respectively).

In both the contra- and the ipsilateral DRG, the majority of FB^+^/PNX^+^/SP^+^ neurons were unevenly distributed, usually as isolated cells scattered throughout the individual ganglionic subdomains. Only a few PNX^+^/SP^+^ UB-ANs were found to form small, loose clusters of up to three neurons. In the contralateral DRG, a distinct accumulation of the retrogradely labelled PNX^+^/SP^+^ neurons was found in the P and the Cd ganglion domains (32.8 ± 2.3% and 31.9 ± 1.9%, respectively), while such labelled UB-ANs were less numerously present in the Md, the Cr and the Cn part of the ganglia (14.1 ± 3.0%, 13.3 ± 4.6% and 7.9 ± 1.6%, respectively). In the ipsilateral ganglia, the majority of PNX^+^ UB-ANs containing SP were located in the Cd and the Cr domains of DRG (36.3 ± 6.9% and 26.7 ± 6.0%, respectively), while a smaller number of retrogradely labelled PNX^+^/SP^+^ sensory neurons was found in the P, the Md and the Cn parts (15.1 ± 4.9%, 14.2 ± 1.8% and 7.7 ± 2.9%, respectively) of all DRG studied. It should be stressed that, in comparison to the contralateral DRG, a distinctly higher number of such labelled UB-ANs was observed in the Cr domain, while in the P part of the ganglion the number of retrogradely labelled PNX^+^/SP^+^ neurons were present in a significantly smaller number. The distribution pattern of PNX^+^ and SP-containing bladder-supplying sensory neurons in the different domains of the DRG studied is showed in Table 5.

#### 3.3.2. Sensory Neurons Co-Localizing PNX and CGRP

The second most numerous subpopulation of FB^+^/PNX^+^ sensory neurons in the investigated DRG was that containing CGRP (50.1 ± 5.4%; Figure 3(B1–B4)) and being present in all the selected contra- and the ipsilateral DRG (51.8 ± 4.1% and 50.2 ± 6.3%) of the L- and the S-Cq-supplying “centers” (48.2 ± 4.9% and 51.4 ± 6.1%). No statistically significant differences were observed in the number of PNX^+^/CGRP^+^ UB-ANs between the ipsi- and contralateral ganglia or between the L and S-Cq DRGs studied.

FB^+^ and PNX/CGRP-containing DRG neurons belonged to two classes of afferent perikarya. The most numerous subpopulation of this type of sensory neurons found in all the DRG studied was the small-sized type (65.8 ± 3.9%), while the medium-sized sensory neurons were found to be distinctly less numerous (Figure 3(B1–B4); 34.2 ± 3.9%). It should be noted that significantly higher number of small-sized neurons and, correspondingly fewer medium-sized neurons were observed in the ipsilateral ganglia (74.5 ± 5.2% and 25.5 ± 5.2%) compared to the contralateral DRG (59.7 ± 3.4% and 40.3 ± 3.4%).

In both the contra- and the ipsilateral DRG, the majority of FB^+^/PNX^+^ neurons simultaneously containing CGRP were unevenly distributed, visible usually as isolated cells, scattered throughout the individual ganglionic subdomains. In the contralateral DRG, a distinct accumulation of the retrogradely labelled PNX^+^/CGRP^+^ sensory neurons was found in the Cd and the P ganglion domains (30.6 ± 2.1% and 29.1 ± 1.5%), while such labelled UB-ANs were less numerously present in the Md, the Cr and the Cn part of the ganglia (16.1 ± 8.7%, 15.4 ± 3.5% and 8.8 ± 1.1%, respectively). In the ipsilateral ganglia, the majority of PNX^+^ UB-ANs containing CGRP were located in the Cd, the Md and the Cr domains of DRG (33.8 ± 3.5%, 24.3 ± 3.6% and 22.9 ± 4.8%, respectively), while a smaller number of retrogradely labelled PNX^+^/CGRP^+^ sensory neurons was found in the Cn and the P parts (11.7 ± 0.8% and 7.3 ± 1.6%) of all DRGs studied. It should be stressed that in comparison to the contralateral DRG, the distinctly higher number of such labelled UB-ANs was observed in the Cn domain of the ipsilateral DRG. The distribution pattern of PNX^+^ and CGRP-containing bladder sensory neurons in the different domains of the studied DRG has is in Table 6.

#### 3.3.3. Sensory Neurons Co-Localizing PNX and GAL

The third most abundant subpopulation of PNX^+^ cells consisted of neurons co-expressing GAL (29.7 ± 2.1%; Figure 3(C1–C4)), present in both the ipsi- and contralateral DRG studied (37.2 ± 5.3% and 52.9 ± 8.65). Although the differences in the numbers of PNX/GAL-IR bladder sensory neurons were not statistically significant between the ipsi- and the contralateral ganglia studied, it should be emphasized that in the S-Cq DRG, a statistically significant higher number of such labelled sensory neurons (44.4 ± 3.2%) was observed when compared with the lumbar ones (21.7 ± 5.3%).

Regarding their diameter, all PNX/GAL-positive UB-ANs belonged exclusively to the subpopulation of small-sized cells (Figure 3(C1–C4)—small arrows).

In both the contra- and ipsilateral DRG, the distribution of retrogradely labelled PNX^+^/GAL^+^ neurons was virtually restricted to only two domains of sensory ganglia: the Cd and the Cr regions (55.8 ± 5.8% vs. 62.1 ± 2.2% and 44.2 ± 5.8% vs. 37.9 ± 2.2%, respectively). This pattern of intraganglionic distribution was repeated in the L as well as the S-Cq UB-ANs (Cd—52.9 ± 3.6% vs. 59.4 ± 1.3% and Cr—47.1 ± 3.6% vs. 40.6 ± 1.3%, respectively). No statistically significant differences in the number of PNX^+^/GAL^+^ UB-ANs were observed in the above-mentioned domains between the ipsi- and contralateral ganglia or between the L and S-Cq DRG studied. The distribution pattern of FB^+^/PNX^+^/GAL^+^ sensory neurons in particular domains of studied ganglia is summarized in Table 7.

#### 3.3.4. Sensory Neurons Co-Localizing PNX and PACAP

PACAP was found in 25.0 ± 1.6% (Figure 3(D1–D4)) of all PNX-containing UB-ANs and it has been present in all selected ipsilateral and contralateral DRG (43.4 ± 3.9% and 38.8 ± 2.6%, respectively). Although the differences in the numbers of PNX/PACAP-IR bladder sensory neurons were statistically insignificant between the ipsi- and the contralateral sensory ganglia, it should be stressed that in the S-Cq DRG, the statistically significant higher number of such labelled sensory neurons (40.9 ± 4.4%) has been observed when compared with the L ones (7.5 ± 0.8%).

PNX-positive UB-ANs containing PACAP belonged mainly to the subpopulation of small-sized cells (Figure 3(D1–D4)-short arrows; 71.3 ± 2.6%), while the number of medium-sized neurons (28.7 ± 2.6%) was distinctly smaller. Although the differences in the numbers of both small PNX/PACAP-IR bladder sensory neurons (63.6 ± 4.6% and 61.6 ± 2.3%) and medium-sized ones (36.4 ± 4.6% and 38.4 ± 2.3%) were not statistically significant between the L and the S-Cq DRG studied, in the ipsilateral ganglia, a noticeably higher number of the small-sized (73.5 ± 3.6%) and a distinctly lower number of medium-sized (26.5 ± 3.6%) PNX- and PACAP-IR FB^+^ neurons were observed when compared with the contralateral ones (44.6 ± 4.9% and 55.4 ± 5.2%).

In terms of the intraganglionic distribution pattern, the majority of FB^+^/PNX^+^ neurons simultaneously containing PACAP were dispersed inside the ganglia, and in both the contra- and the ipsilateral DRG, the largest number of such encoded sensory neurons was observed in the Cd region (42.9 ± 2.3% vs. 36.5 ± 9.0%) of the ganglion. For the remaining DRG domains, in the contralateral DRG, PNX/PACAP-positive UB-ANs were less numerous in the Md, the P, the Cr and the Cn parts of the ganglia (24.6 ± 3.3%, 15.5 ± 1.3%, 11.4 ± 3.3% and 5.6 ± 4.8%, respectively). In the ipsilateral ganglia, a smaller number of retrogradely labelled PNX^+^/PACAP^+^ sensory neurons was found in the Cr, the Md, the P and the Cn parts (23.7 ± 6.0%, 16.6 ± 3.2%, 13.7 ± 2.8% and 9.5 ± 1.9%, respectively) of all the DRG studied. It should be stressed that in comparison to the contralateral DRG, a distinctly smaller number of such labelled UB-ANs was observed in the P and the Md domain of the ipsilateral DRG. The distribution pattern of PNX^+^ and PACAP-containing bladder sensory neurons in the different domains of the DRGs investigated is summarized in Table 8.

#### 3.3.5. Sensory Neurons Co-Localizing PNX and nNOS

PNX^+^/nNOS-containing sensory neurons constituted one of the smallest subpopulations of all PNX-IR and retrogradely traced neurons in DRG (14.9 ± 2.1%; Figure 3(E1–E4)) and were observed in all the ipsi- and the contralateral DRG studied, constituting 15.9 ± 5.9% and 6.8 ± 2.5% of all PNX-positive UB-ANs, respectively. Additionally, the L DRG contained a significantly higher number of PNX/nNOS-positive retrogradely traced neurons (16.4 ± 2.4%) than S-Cq ganglia (7.7 ± 3.0%).

In general, nNOS was observed in two size classes of PNX^+^ afferent perikarya, being present in 58.4 ± 7.7% of small- and 41.6 ± 7.7% of medium-sized sensory neurons, respectively. Although the differences in the numbers of both small PNX/nNOS-IR bladder sensory neurons (Figure 3(E1–E4)-short arrows) and medium-sized ones were statistically insignificant between the L (64.6 ± 5.0% and 52.7 ± 6.3%) and the S-Cq (35.4 ± 5.0% and 47.3 ± 6.3%) DRGs studied, in the ipsilateral ganglia the statistically significant higher number of the small-sized neurons (63.5 ± 4.2%) and the distinctly lower number of medium-sized (36.5 ± 4.2%) PNX and nNOS-IR FB^+^ neurons were observed when compared with the contralateral ones (53.5 ± 3.9% and 46.5 ± 3.9%, respectively).

In the ipsilateral ganglia, PNX^+^/nNOS^+^ UB-ANs were dispersed inside the ganglia, mainly in the Cd (32.6 ± 2.7%) and in the Md (30.3 ± 2.7%) regions. The remainder of nNOS-containing, PNX^+^ sensory neurons were present in the Cr, Cn and P domains of the DRG studied (15.9 ± 1.4%, 11.1 ± 3.7% and 10.1 ± 5.7%, respectively). In the contralateral DRG, the distribution of retrogradely labelled PNX^+^/nNOS^+^ neurons were virtually restricted to only two domains of sensory ganglia: the Cd and the Md regions (79.9 ± 7.6% and 20.1 ± 7.9%, respectively). It should be stressed that in comparison to the contralateral DRG, in the ipsilateral ganglia, a distinctly higher number of such labelled UB-ANs was observed in the Cn and P domains (11.1 ± 3.7% vs. 0.0% and 10.1 ± 5.7% vs. 0.0%). In contrast, in the Cd part of the ganglion, the number of retrogradely labelled PNX^+^/nNOS^+^ sensory neurons was significantly smaller (32.6 ± 2.7% vs. 79.9 ± 7.6%). Distribution pattern of PNX^+^ and nNOS-containing bladder sensory neurons in the different domains of DRG is shown in Table 9.

#### 3.3.6. Sensory Neurons Co-Localizing PNX and CRT

The presence of PNX and CRT was demonstrated in a small population (6.5 ± 0.3%; Figure 3(F1–F4)) of sensory DRGs supplying the urinary bladder. No statistically significant differences in the numbers of FB^+^/PNX^+^/CRT^+^ neurons were observed either between the ipsi- and contralateral DRGs studied (6.5 ± 0.7% and 9.4 ± 3.4%) or between the L and S-Cq bladder-supplying “centers” (6.4 ± 1.6% and 7.9 ± 3.4%).

Regarding the diameter, the most numerous subpopulation of PNX-positive UB-ANs containing CRT was that of small-sized neurons (Figure 3(F1–F4); 75.6 ± 1.6%), while the medium-sized neurons were found to be distinctly less numerous (24.4 ± 1.6%). It should be stressed that the differences in the numbers of both small PNX/CRT-IR bladder sensory neurons (71.1 ± 8.2% vs. 77.5 ± 1.1%) and those of the medium-sized neurons (28.9 ± 8.2% vs. 22.5 ± 1.1%) were not statistically significant between the L and the S-Cq DRG and between the ipsi- and contralateral ones (80.5 ± 7.8% vs. 66.7 ± 5.4% and 19.5 ± 7.8% vs. 33.3 ± 5.4%, respectively).

In terms of the intraganglionic distribution pattern, the majority of FB^+^/PNX^+^/CRT^+^ sensory neurons were unevenly dispersed inside the ganglia; the largest number of such encoded neurons was observed in the Cd region (96.3 ± 3.7% vs. 54.7 ± 10.8%) in both the ipsi- and the contralateral ganglia. For the remaining DRG domains, in the contralateral DRG, the rest of PNX/CRT-positive UB-ANs were only present in the Md part of the ganglion (3.7 ± 3.7%). In the ipsilateral ganglia, a smaller number of retrogradely labelled PNX^+^/CRT^+^ sensory neurons was found in the Md, P and Cn parts (29.6 ± 4.5%, 10.5 ± 9.1% and 5.2 ± 4.5%, respectively) of all DRG studied. It should be stressed that in comparison to the contralateral DRG, in the ipsilateral ganglia, a distinctly higher number of such labelled UB-ANs was observed in the Md domain, while the retrogradely labelled PNX^+^/CRT^+^ sensory neurons were present in a significantly smaller number in the Cd domain. The distribution pattern of PNX^+^ and CRT-containing bladder-projecting sensory neurons in the different domains of DRGs is presented in Table 10.

#### 3.3.7. Sensory Neurons Co-Localizing PNX and SOM

The SOM-containing sensory neurons constituted the smallest subpopulation (1.6 ± 1.1%; Figure 3(G1–G4)) of all PNX^+^ UB-ANs, being observed in all the ipsi- and contralateral DRG (1.5 ± 1.2% and 1.7 ± 1.6%, respectively) of the L and the S-Cq “centers” (1.0 ± 0.7% and 1.5 ± 1.8%, respectively). No statistically significant differences in the number of PNX^+^/SOM^+^ UB-ANs were observed between the ipsi- and contralateral ganglia and the L and S-Cq DRG studied.

The PNX-positive, SOM-containing UB-ANs belonged mainly to the subpopulation of small-sized cells (Figure 3(G1–G4), short arrows; 83.1 ± 2.6%), while the number of such labelled medium-sized neurons was distinctly smaller (16.9 ± 2.6%). Although the differences in the numbers of both small PNX/SOM-IR bladder sensory neurons (74.8 ± 4.8% vs. 93.7 ± 5.6%) and medium-sized ones (25.2 ± 4.8% vs. 6.3 ± 5.6%) were not statistically significant between the ipsi- and the contralateral DRG studied, in the S-Cq ganglia, the noticeably higher number of the small-sized (96.3 ± 3.6% vs. 69.4 ± 2.7%) and the distinctly lower number of medium-sized (3.7 ± 3.6% vs. 30.6 ± 2.7%) PNX and SOM-IR FB^+^ neurons were observed when compared with the L ones.

In terms of the intraganglionic distribution pattern, in both the contralateral and the ipsilateral DRG, the majority of FB^+^/PNX^+^/SOM^+^ neurons were observed in the Cd region (95.9 ± 5.5% and 55.7 ± 8.0%, respectively) of the ganglia. For the remaining DRG domains, the rest of PNX/SOM-positive UB-ANs were located in the P segment of the contralateral DRG (4.1 ± 5.5%), while in the ipsilateral ganglia, a smaller number of retrogradely labelled PNX^+^/SOM^+^ sensory neurons was found in the Cr and the P regions (32.8 ± 4.0% and 11.5 ± 5.1%, respectively) of studied DRG. It should be stressed that in comparison to the contralateral DRG, in the ipsilateral ganglia, a distinctly higher number of such labelled UB-ANs was observed in the Cr domain (32.8 ± 4.0% vs. 0.0%), while a significantly smaller number of retrogradely labelled PNX^+^/SOM^+^ sensory neurons (55.7 ± 8.0% vs. 95.9 ± 5.5%) was present in the Cd part of the ganglion. The distribution pattern of PNX^+^ and SOM-containing bladder sensory neurons within different domains of DRGs is summarized in Table 11.

### 3.4. Pattern of Co-Occurrence of Biologically Active Substances in the PNX-Containing UB-ANs of DRG Studied

An attempt to colocalize more than two studied substances within the same neuron profile, using consecutive sections labelled by the double immunofluorescence technique, allowed us to draw the following conclusions:(i)Almost half of the PNX^+^/SP^+^ UB-ANs co-contain PACAP, CGRP or GAL (50.6 ± 4.5%, 46.5 ± 0.9% and 45.9 ± 1.9%, respectively). A much smaller number of such labelled FB-positive sensory neurons contained CRT, nNOS or SOM (9.2 ± 1.9%, 3.8 ± 1.4% and 1.2 ± 1.1%, respectively).(ii)SP, PACAP and GAL were also very frequently observed (69.8 ± 1.6%, 49.7 ± 2.2% and 47.2 ± 0.9%, respectively) in the PNX^+^/CGRP^+^ sensory neurons supplying the urinary bladder, while PNX^+^/CGRP^+^ neurons simultaneously containing nNOS, CRT and/or SOM (9.8 ± 1.9%, 8.4 ± 3.6% and 1.4 ± 1.4%, respectively) were observed much less frequently.(iii)The most numerous subset of PNX^+^/GAL^+^ UB-ANs was that simultaneously containing SP, CGRP and/or PACAP (68.2 ± 7.4%, 60.1 ± 7.2% and 49.4 ± 2.7%, respectively). In contrast, FB-positive neurons containing both PNX and GAL rarely colocalized nNOS, CRT or SOM (8.4 ± 2.8%, 8.1 ± 0.6% and 1.9 ± 0.8%, respectively).(iv)A similar pattern of colocalization was observed for PNX and PACAP-positive UB-ANs, where the vast majority of such encoded sensory neurons contained SP, CGRP and/or GAL (82.1 ± 4.5%, 61.1 ± 2.0% and 49.2 ± 5.1%, respectively), while such labelled neurons, showing immunoreactivity for nNOS, CRT and/or SOM, were observed in much smaller numbers (15.5 ± 5.9%, 8.8 ± 0.3% and 1.9 ± 0.8%, respectively).(v)The most numerous subpopulations of FB^+^/PNX^+^/NOS^+^ sensory neurons were those that simultaneously contained PACAP, CGRP and/or SP (79.2 ± 12.0%, 64.4 ± 2.0% and 57.9 ± 2.7% respectively). GAL and CRT were observed in a much smaller subset (40.9 ± 2.7% and 25.1 ± 8.9) of these neurons, while SOM was absent from this subpopulation of UB-ANs.(vi)PNX-IR, CRT-positive neurons mainly co-contained SP and CGRP (79.1 ± 1.5%, 54.3 ± 6.9%, respectively), while PACAP, GAL and/or nNOS (39.5 ± 7.3%, 37.8 ± 3.6% and 23.2 ± 3.3%) were observed in a much smaller population of FB^+^/PNX^+^/CRT^+^ perikarya. As in the above-mentioned case, SOM was absent from this subpopulation of UB-ANs.(vii)All PNX/SOM-positive UB-ANs simultaneously contained SP and CGRP, while more than half of these neurons colocalized with GAL and PACAP (70.3 ± 4.1% and 68.9 ± 5.5%). This population of sensory neurons was devoid of immunoreactivities towards nNOS and CRT.

Mathematical analysis of the relative frequencies of co-occurring antigens studied in cross-sections of the same sensory neuron, analyzed on consecutive sections of the ganglion, indicates the existence of two basic populations of afferent neurons supplying the urinary bladder. The basic determinant of both populations is the presence (minority, less than 2% of all perikarya) or absence of SOM expression (the overwhelming majority of the labeled neurons studied). While the population characterized by the presence of SOM is very homogeneous in terms of chemical coding (in fact, the vast majority of these sensory neurons simultaneously express SP, CGRP, SOM, PACAP and GAL (approximately 70% of neurons in this population), the second main population of UB-ANs, as the results from the aforementioned mathematical analysis indicate, consists of multiple fractions of DRG neurons with different chemical encoding.

Although the vast majority of sensory neurons of the latter subpopulation are small-diameter cells, usually identifiable on a maximum of three consecutive sections, the present study could not provide indisputable evidence for the presence of the cell populations mentioned below; however, the analysis of the number of cells expressing individual antigens allows us to state that these sensory cells can be classified as neurons “coded” by the simultaneous presence of SP/CGRP/PACAP/GAL, SP/CGRP/PACAP/NOS, SP/CGRP/PACAP/NOS/CRT and/or SP/CGRP/GAL/PACAP.

Details concerning the relative percentages of individual subpopulations of retrogradely labeled PNX^+^ neurons are shown in Table 12.

## 4. Discussion

The analysis of the literature clearly shows that PNX may be involved in sensory transmission. PNX was detected in the epidermis and dermis; moreover, this peptide was found also in the skin-supplying DRG neurons [16]. It has also been demonstrated that PNX exhibits a distribution in the dorsal horn and DRGs similar to that of gastrin-releasing peptide, which causes itching [16].

### 4.1. Distribution Pattern and Morphometrical Characteristics of PNX-Containing UB-ANs

Lyu et al. revealed that PNX is expressed in DRG neurons in rodents [5]; however the results of the present study show, for the first time, that PNX is present in the porcine DRG sensory neurons supplying the urinary bladder. This neuropeptide was detected in the small- and medium-sized DRGs, but it was absent from the large-diameter cells. It is well known that the small- and medium-sized DRG neurons are involved in pain transmission [17,18]. Considering that Lyu et al. showed that PNX can reduce visceral pain, but not thermally evoked pain [5], this phenomenon may also suggest the putative influence of PNX on nociceptive transmission from the urinary bladder.

### 4.2. Sensory Neurons Co-Localizing PNX and SP

The main co-transmitter of PNX is SP, a tachykinin distributed throughout the central and the peripheral nervous systems. The literature clearly shows that SP is involved in the classical and neurogenic inflammatory response at the peripheral level [19,20]. In the lower urinary tract, SP induces a bladder contraction and facilitates normal micturition [21]. Moreover, SP is recognized as a peptide involved in the nociceptive transmission [22]. Co-expression of PNX with SP may suggest that PNX, being present in a significant fraction of SP-IR sensory neurons, influences the micturition and the nociception, by modulation of SP-driven stimulation. However, this speculation requires confirmation in further studies.

### 4.3. Sensory Neurons Co-Localizing PNX and CGRP

CGRP is a peptide associated with several pathological conditions, including the modulation of inflammatory cytokine release during sepsis and in the airways through hyperemia [23]. It was also shown that CGRP levels increase during migraine attacks [24]. Considering the above data, as well as the relatively frequent co-occurrence with SP in retrogradely labeled DRG cells, it may be speculated that CGRP is also involved in nociceptive transmission. Although in the lower urinary tract CGRP has no excitatory effect on the micturition reflex pathway per se, it is able to facilitate the SP-evoked chemonociceptive reflex [25]. CGRP inhibits the activity of an endopeptidase that degrades SP [26], and thus may “improve” and extend the time of SP action. A large number of FB^+^ PNX-IR neurons containing SP and/or CGRP indicates the putative connection of PNX with bladder contraction and nociceptive transmission. Moreover, it is widely known that the release of CGRP and SP increases significantly following inflammation [27]. This strongly suggests that PNX may involve sensory transmission.

### 4.4. Sensory Neurons Co-Localizing PNX and PACAP

The next most abundant population of PNX^+^ sensory neurons, as shown by double immunofluorescence studies, were neurons simultaneously containing either PACAP or GAL. The presence of PACAP in sensory neurons serving various afferent pathways has been found in virtually every area of the human body [28], and this molecule seems to play a significant role in many different physio- and pathophysiological processes. For example, it was investigated and observed that the intravenous infusion of PACAP in healthy human volunteers caused a slight headache or, rather, a feeling of increasing pressure in the head [29], which may suggest that, similarly to CGRP, this transmitter may also be a factor involved in the etiopathogenesis of migraine attacks. This is further supported by the observations that in rats, administration of pituitary adenylate cyclase-activating polypeptide type I receptor (PAC1) antagonists both blocks the development of migraine and significantly reduces nociceptive transmission itself [30]. Moreover, PACAP has been shown to act as both a significant anti-inflammatory and neuroprotective factor in brain tissue [31], acting via stimulation and recovery of brain-derived neurotrophic factor (BDNF) expression under pathophysiological conditions [32,33]. In the lower urinary tract, PACAP increased bladder smooth muscle tone and potentiated electric field stimulation (EFS)-induced contractions [34]. On the other hand, Hernández et al. reported that PACAP induced relaxations of the pig urinary bladder neck [35]. Considering the above-mentioned data, it can be assumed that PACAP plays a number of roles in the lower urinary tract, starting from being involved in the regulation of the detrusor muscle tone and the control of blood flow through the vascular bed of the organ, to acting as a sensory transmitter and, probably, as a neuroprotective factor.

### 4.5. Sensory Neurons Co-Localizing PNX and GAL

It is well-known that GAL is involved in the regulation of many processes within the lower urinary tract. As can be concluded from the available literature, this peptide seems to influence the contractility of the detrusor muscle in the rat bladder per se; moreover, it may also act indirectly, modulating the activity of autonomic ganglia neurons involved in the control of detrusor motility [36]. It seems that the action of GAL in the lower urinary tract is not limited to the effect on smooth muscles; rather, through its ability to modulate the activity of neuromuscular junctions [37], it may also (co-)regulate the activity of the external urethral sphincter. Furthermore, Honda and colleagues showed that intrathecal administration of this peptide delays the onset of micturition in rats, which may be indicative of an inhibitory role of GAL in the control of micturition reflex at the spinal level [38]. It is also interesting that GAL may interact with other known afferent cell transmitters; for example, Zvarova et al. reported that GAL may inhibit the actions exerted by PACAP and/or nitric oxide (NO) on bladder afferent cells [39]. Furthermore, GAL, when administrated intrathecally in higher doses, blocked the facilitatory effects of SP and CGRP on the excitability of the nociceptive flexor reflex in rats [40,41], implying that GAL may possess both analgesic and anti-inflammatory properties [42]. Considering that PNX was colocalized with GAL in a fair number of neurons, it seems likely that PNX may participate in the regulatory effects described above by modulating the GAL function in a currently unknown manner. However, the existence or absence of such interactions remains to be clarified in detail. It should be emphasized that PNX co-occurs in the DRGs supplying the bladder with transmitters that are believed to induce opposing physiological effects (e.g., SP and CGRP facilitate micturition and nociceptive transmission, while GAL inhibits these phenomena), and further studies are necessary to explain the interactions between PNX and other active substances co-occurring with it in sensory neurons.

In contrast to the substances described above, PNX co-existed with nNOS, CRT and/or SOM only in a marginal number of retrogradely labeled cells.

### 4.6. Sensory Neurons Co-Localizing PNX and nNOS

The role of NO in regulating the lower urinary tract function, particularly the pool of molecules released from afferent nerve endings supplying the wall of the bladder and/or urethra, has not yet been sufficiently explained. On the one hand, it has been shown that the major sites of NO release in the rat bladder are afferent nerves and that NO plays an important role in the facilitation of the micturition reflex evoked by noxious chemical irritation of the bladder [43], simultaneously participating in the initiation of inflammatory responses and triggering painful sensations [43,44]. On the other hand, an inhibitory function of NO in the micturition pathway has also been suggested. It was reported that NO may be a regulator of sodium currents in C-type DRGs, leading, by suppression of both fast and slow sodium trafficking, to DRG hypoexcitability [25]. As previously described in the rat, similarly to PNX, nNOS is expressed mainly by small-diameter sensory neurons [45]; moreover, this enzyme co-expresses very frequently with SP and/or CGRP in L DRGs [46]; however, in the present study, only a small percentage of retrogradely labeled PNX^+^ sensory neurons co-expressed nNOS, even though a significant proportion of these neurons co-expressed SP and/or CGRP. This may suggest that the modulatory effect of PNX on nitrergic pathways regulating bladder functions is rather limited; however, considering the frequency of co-existence of PNX with SP and CGRP, its role seems to be much more important in the latter case.

### 4.7. Sensory Neurons Co-Localizing PNX and SOM

At present, the physiological significance of SOM in the bladder afferent neurons remains completely unclear. The only hypothesis that can be derived without any doubt from the presented results, analyzing both the number of cells containing it and the patterns of colocalization of this transmitter with other biologically active substances studied in this experiment, is that it most likely constitutes a distinct component of the bladder afferent pathways, the significance of which must be clarified in future studies. At present, it can only be assumed, based on the available research results, that SOM may be involved in the specific modulation of nociceptive transmission [47], depressing the firing of dorsal horn neurons activated by noxious stimulation [48], which is confirmed by the observations that SOM may produce an antinociceptive effect [49], especially in the case of inflammatory reactions of both neurogenic and non-neurogenic origin [50].

### 4.8. Sensory Neurons Co-Localizing PNX and CRT

CRT was the last protein that was previously observed by us in the DRGs associated with the bladder innervation and whose co-occurrence with PNX we decided to investigate in this study. Although the existence of CRT was already described in 1987 [51], to date, there are no precise data allowing to strongly suggest its physiological function in the DRGs involved in the regulatory loops of the micturition process. While it is known that CRT is a protein efficiently modulating both transmembrane Ca^2+^ currents and calcium homeostasis in neurons of many parts of brain (e.g., hippocampus, thalamus, neocortex, amygdala) as well as spinal cord regions (especially in the superficial laminae of dorsal horn) [52,53], the exact role of this protein in DRG sensory neurons remains to be elucidated. Although the ability to modify the presynaptic signaling and Ca^2+^ transients by CRT acts as a buffer against excitotoxicity, the expression of this protein seems to be modulated by hypoxic stress, as observed after ischemic episodes [54], since its overexpression was observed in neurons belonging to injured brain areas [55]. Another interesting observation is that the mechanical and chemical stimuli received and transmitted by DRGs induce the expression of CRT in the sensory neurons of dorsal horns of the spinal cord, as can be judged by the increase in the number of CRT^+^ cells visible in the spinal cord sections within the superficial Rexted laminae, a phenomenon that was explained by the increased activity of nociceptive transmission under the influence of the above-mentioned stimuli [56]. However, further dedicated studies are required to reliably establish the role of this calcium-binding protein and its (possible) interactions with PNX and other studied compounds in the regulation of bladder function.

### 4.9. Pattern of Co-Occurrence of Biologically Active Substances in the PNX-Containing UB-ANs

In the course of this study, we also tried to answer the question whether PNX^+^ DRG afferent cells, involved in the innervation of the urinary bladder, can simultaneously colocalize more than two tested substances. However, although the analysis of the co-occurrence patterns of the tested substances, performed on consecutive sections, provided a substantial amount of new data (see Table 12), expanding our knowledge of the chemical coding of bladder sensory neurons, it also raised almost as many new questions, the answers to which require in-depth studies, both qualitative and functional. Although the existence of two clearly separable subpopulations of bladder sensory neurons, distinguished by the expression (or lack thereof) of SOM, seems indisputable, the physiological significance of this subtype of sensory perikaryons and the function(s) that may be assigned to them remain unknown at present. When the chemical coding patterns of the second, much larger subpopulation of PNX^+^ DRG cells conducting sensory information from the bladder wall are analyzed, it is clear that their suite of neurotransmitters represents variant patterns of coexistence of GAL, PACAP, nNOS and/or CRT with the “core pair” of sensory transmitters, SP and/or CGRP. While the importance of the co-occurrence and simultaneous or sequential release of two neurotransmitters from sensory neurons into the target tissue has been investigated in a number of studies, there are no similar studies on both the physiological importance and the interrelationships involved in the simultaneous release of more types of neurotransmitter molecules from the same nerve ending into the target tissue. An additional difficulty in interpreting the functional significance of the observed compositions of active substances that may be released into target tissues is both the almost complete ignorance of the latter (in relation to the individual neuronal subtypes identified during these studies) and the lack of data on the receptor “landscape” of potential effector tissues. Unfortunately, it should be stated that as of yet, without further, in-depth functional studies, the multitude of observed patterns of co-occurrence of the transmitters studied makes it impossible to clearly assign individual sensory modalities to individual sub-types of differently coded DRGs.

### 4.10. Future Perspective and Clinical Implications

The above-described experimental work provided data that, for the first time, conclusively demonstrate not only the presence of PNX in the porcine DRG sensory neurons supplying the urinary bladder but also its coexistence pattern with other neurotransmitters in the examined UB-ANs. This information may be helpful in improving our understanding the functional role of PNX in the transmission of various types of sensory information and its possible impact on the function of the urinary bladder. Additionally, since many urinary bladder disorders have a neurogenic background, better understanding the nervous mechanisms controlling the proper micturition and function of this organ will allow for a better understanding of the pathological mechanisms causing dysfunctions of the urinary bladder and enable more precise treatment of the neurogenic bladder disorders.

## 5. Conclusions

The following conclusions can be drawn from the results of this study presented:(i)PNX occurs in a relatively large population of DRG neurons innervating the porcine urinary bladder, which indicates its participation in regulating and/or modulating the activity of afferent neural pathways co-creating the regulatory loops of micturition and/or reception and transmission of various sensory modalities from the organ wall;(ii)The surprisingly numerous patterns of co-occurrence of PNX and many substances considered to be sensory transmitters strongly suggest that this peptide may be involved in a wide range of mechanisms regulating the physiological behavior of the organ;(iii)The multitude of observed patterns of co-occurrence of PNX and other tested transmitters/biologically active substances necessitates further, in-depth studies focusing on the detailed determination of target tissues, the expression patterns of types and subtypes of receptors, and the undisputed indication of the functions performed by individual subsets of differently coded DRG cells.

## Figures and Tables

**Figure 1 cells-14-00516-f001:**
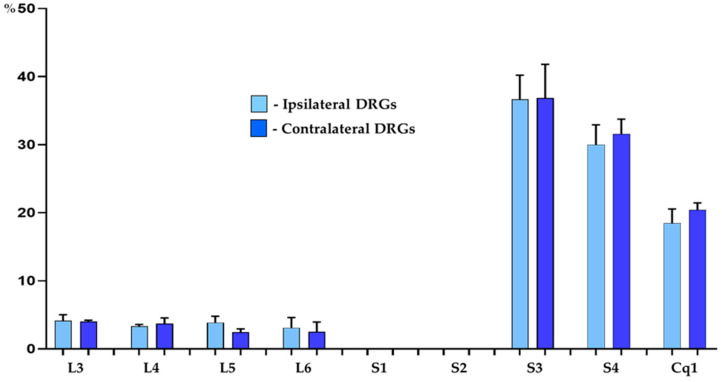
Bar diagram showing the distribution pattern of FB^+^ bladder sensory neurons in the porcine DRG ipsilateral (light blue bars) and contralateral (dark blue bars) to the sites of tracer injections into the organ wall. The data obtained were pooled in all the animals and presented as mean ± SD, with *N* = 6 animals.

**Figure 2 cells-14-00516-f002:**
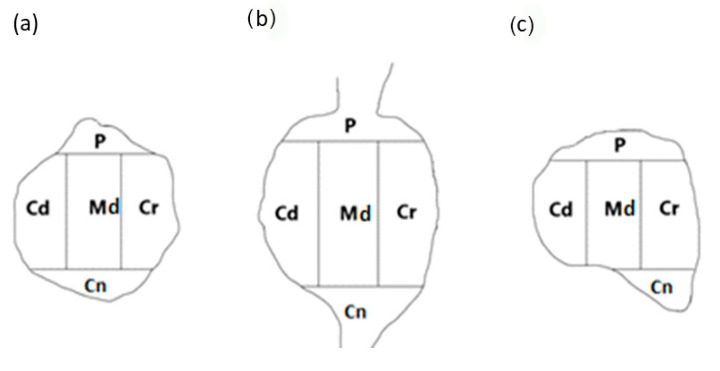
A schematic diagram of a DRG section showing its arbitrary division into topographical domains, in which the occurrence and relative frequency of PNX-containing sensory neurons was studied: P—peripheral domain, Cr—cranial domain, Cd—caudal domain, Cn—central domain of the DRG, Md—middle ganglion area; section from the (**a**) proximal, (**b**) middle and (**c**) distal part of the ganglion. The same mask was used in studies on the intraganglionic distribution of different chemically coded subpopulations of DRG neurons in previous studies [6].

**Figure 3 cells-14-00516-f003:**
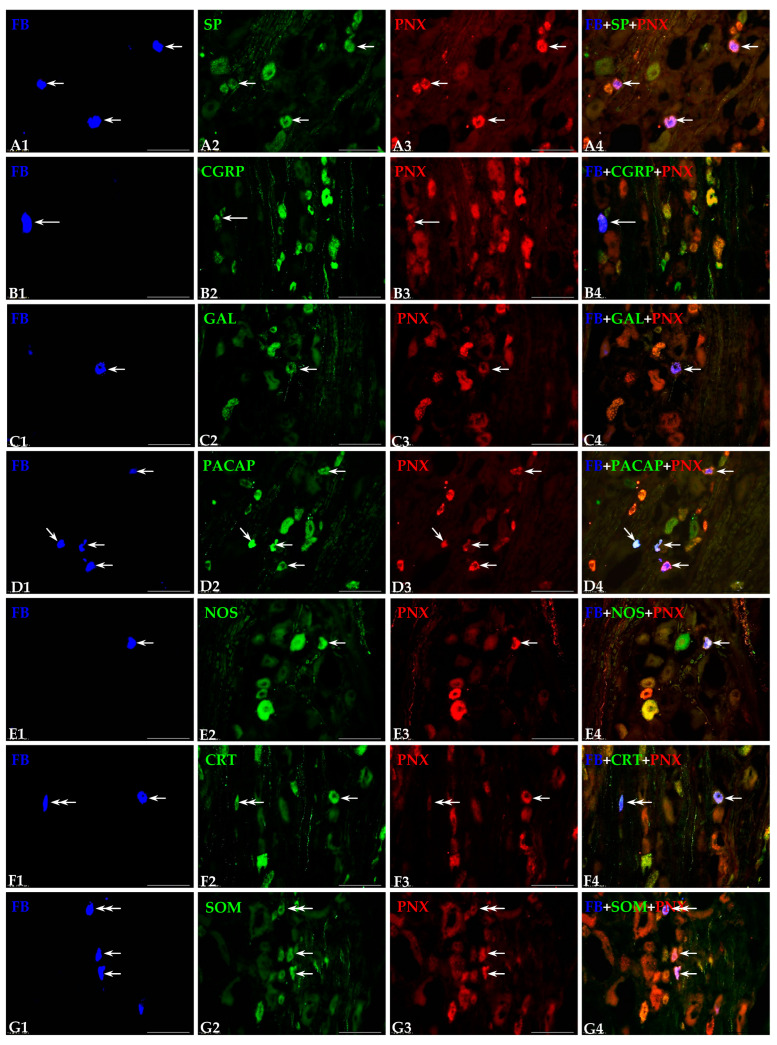
Representative images of dorsal root ganglia (DRG)-UB-ANs. The images were taken separately from blue (**A1**,**B1**,**C1**,**D1**,**E1**,**F1**,**G1**), green (**A2**,**B2**,**C2**,**D2**,**E2**,**F2**,**G2**) and red (**A3**,**B3**,**C3**,**D3**,**E3**,**F3**,**G3**) fluorescent channels. Pictures (**A4**,**B4**,**C4**,**D4**,**E4**,**F4**,**G4**) represent images in which the blue, green and red channels were digitally superimposed. The short arrows represent small-sized FB-positive DRG-UB-ANs (**A1**,**C1**,**D1**,**E1**,**F1**,**G1**) that were simultaneously PNX—(**A3**,**C3**,**D3**,**E3**,**F3**,**G3**) and substance P—(SP; **A2**), galanin—(GAL; **C2**), pituitary adenylate cyclase-activating polypeptide—(PACAP; **D2**), neuronal nitric oxide synthase—(nNOS; **E2**), calretinin—(CRT; **F2**) or somatostatin-positive—(SOM; **G2**). Long arrow represents middle-sized FB-positive DRG-UB-AN (**B1**) containing PNX (**B3**) and CGRP (**B2**). Double arrows indicate small-sized FB-positive DRG-UB-ANs (**F1**,**G1**) that contained CRT (**F2**) or SOM (**G2**) and were simultaneously PNX-negative (**F3**,**G3**). Scale bar in all the images—100 μm.

**Table 1 cells-14-00516-t001:** List of primary antisera and secondary reagents used in this study: CGRP—calcitonin gene-related peptide, CRT—calretinin, GAL—galanin, nNOS—neuronal nitric oxide synthase, PACAP—pituitary adenylate synthase-activating polypeptide, PNX—phoenixin, SOM—somatostatin, SP—substance P, FITC—fluorescein isothiocyanate, CY3—streptavidin-conjugated.

Antigen	Code	Dilution	Host	Supplier
Primary antibodies				
CGRP	PC205L	1:9000	Rabbit	Merck Millipore, Temecula, CA, USA
CRT	6B3	1:2000	Mouse	SWANT, Burgdorf, Switzerland
GAL	AB5909	1:4000	Rabbit	Merck Millipore, Temecula, CA, USA
nNOS	N2280	1:200	Mouse	Sigma-Aldrich, St. Louis, MO, USA
PACAP	T-4465	1:15,000	Rabbit	Peninsula, San Carlos, CA, USA,
PNX	H-079-01	1:7000	Rabbit	Phoenix Pharmaceuticals Inc., Burlingame, CA, USA
SOM	MAB 354	1:50	Rat	Merck Millipore, Temecula, CA, USA
SP	8450-0004	1:200	Rat	Bio-Rad, Kidlington, UK
**Secondary reagents**				
Biotinylated anti-rabbit immunoglobulins	E 0432	1:1000	Goat	Dako, Hamburg, Germany
CY3-conjugated streptavidin	711-165-152	1:12,000	-	Jackson I.R., West Grove, PA, USA
FITC-conjugated anti-rat IgG	712-095-150	1:400	Donkey	Jackson I.R., West Grove, PA, USA
FITC-conjugated anti-mouse IgG	715-096-151	1:600	Donkey	Jackson I.R., West Grove, PA, USA

**Table 2 cells-14-00516-t002:** List of antigens used in pre-absorption test.

Antigen	Code	Dilution	Supplier
CGRP	T-4030	1:800	Peninsula Laboratories, San Carlos, CA, USA
CRT	Lot No.: 22	1:2000	SWANT, Burgdorf, Switzerland
GAL	T-4862	1:1500	Peninsula Laboratories, San Carlos, CA, USA
nNOS	N3033	1:200	Sigma, St. Louis, MO, USA
PACAP	A9808	1:1000	Sigma, St. Louis, MO, USA
PNX	079-01	1:7000	Phoenix Pharmaceuticals Inc., Burlingame, CA, USA
SOM	S9129	1:50	Sigma-Aldrich, St. Louis, MO, USA
SP	S6883	1:200	Sigma-Aldrich, St. Louis, MO, USA

**Table 3 cells-14-00516-t003:** Percentages of Fast Blue-positive (FB^+^) neurons located in the individual lumbar (L), sacral (S) and coccygeal (Cq) dorsal root ganglia (DRG). The data obtained were pooled in all the animals and presented as mean ± standard deviation (SD), with *N* = 6 animals.

FB^+^ Neurons	L3 %	L4 %	L5 %	L6 %	S3 %	S4 %	Cq1 %
Ipsilateral ss	3.5 ± 0.4	2.4 ± 0.4	3.4 ± 0.5	2.9 ± 0.8	33.3 ± 2.0	30.7 ± 1.6	23.8 ± 1.2
Contralateral DRGs	3.8 ± 0.1	3.0 ± 0.1	3.7 ± 0.2	3.2 ± 0.8	33.5 ± 2.8	31.3 ± 1.2	21.5 ± 0.6

**Table 4 cells-14-00516-t004:** Percentages of the intraganglionic distribution pattern of FB^+^/PNX^+^ neuronal populations located in the ipsilateral and the contralateral DRG studied. Data obtained from all studied animals were pooled and presented as mean ± SD, referring to the number of animals (*N* = 6).

DRG Subdomain	P	Cr	Cd	Cn	Md
Ipsilateral DRGs	12.2 ± 2.1%	20.2 ± 4.2%	36.7 ± 3.6%	8.6 ± 0.8%	22.3 ± 1.6%
Contralateral DRGs	24.8 ± 1.6%	13.2 ± 1.2%	26.5 ± 1.5%	17.4 ± 1.4%	18.1 ± 4.3%

**Table 5 cells-14-00516-t005:** Percentages of the intraganglionic distribution pattern of FB^+^/PNX^+^ neuronal populations containing SP and located in the ipsilateral and the contralateral DRG studied. Data obtained from all studied animals were pooled and presented as mean ± SD, with *N* = 6 animals.

DRG Subdomain	P	Cr	Cd	Cn	Md
Ipsilateral DRGs	15.1 ± 4.9%	26.7 ± 6.0%	36.3 ± 6.9%	7.7 ± 2.9%	14.2 ± 1.8%
Contralateral DRGs	32.8 ± 2.3%	13.3 ± 4.6%	31.9 ± 1.9%	7.9 ± 1.6%	14.1 ± 3.0%

**Table 6 cells-14-00516-t006:** Percentages of the intraganglionic distribution pattern of FB^+^/PNX^+^ neuronal populations containing CGRP and located in the ipsi- and contralateral DRG Data obtained from all studied animals were pooled and presented as mean ± SD, with *N* = 6 animals.

DRG Subdomain	P	Cr	Cd	Cn	Md
Ipsilateral DRGs	7.3 ± 1.6%	22.9 ± 4.8%	33.8 ± 3.5%	11.7 ± 0.8%	24.3 ± 3.6%
Contralateral DRGs	29.1 ± 1.5%	15.4 ± 3.5%	30.6 ± 2.1%	8.8 ± 1.1%	16.1 ± 8.7%

**Table 7 cells-14-00516-t007:** Percentages of the intraganglionic distribution pattern of FB^+^/PNX^+^ neuronal populations containing GAL and located in the ipsi- and contralateral DRG studied. Data obtained from all studied animals were pooled and presented as mean ± SD, with *N* = 6 animals.

DRG Subdomain	P	Cr	Cd	Cn	Md
Ipsilateral DRGs	0%	37.9 ± 2.2%	62.1 ± 2.2%	0%	0%
Contralateral DRGs	0%	44.2 ± 5.8%	55.8 ± 5.8%	0%	0%

**Table 8 cells-14-00516-t008:** Percentages of the intraganglionic distribution pattern of FB^+^/PNX^+^ neuronal populations containing PACAP and located in the ipsi- and contralateral DRG studied. Data obtained from all studied animals were pooled and presented as mean ± SD, with *N* = 6.

DRG Subdomain	P	Cr	Cd	Cn	Md
Ipsilateral DRGs	13.7 ± 2.8%	23.7 ± 6.0%	36.5 ± 9.0%	9.5 ± 1.9%	16.6 ± 3.2%
Contralateral DRGs	15.5 ± 1.3%	11.4 ± 3.3%	42.9 ± 2.3%	5.6 ± 4.8%	24.6 ± 3.3%

**Table 9 cells-14-00516-t009:** Percentages of the intraganglionic distribution pattern of FB^+^/PNX^+^ neuronal populations containing nNOS and located in the ipsilateral and contralateral DRG studied. The data obtained from all studied animals were pooled and presented as mean ± SD, with *N* = 6 animals.

DRG Subdomain	P	Cr	Cd	Cn	Md
Ipsilateral DRGs	10.1 ± 5.7%	15.9 ± 1.4%	32.6 ± 2.7%	11.1 ± 3.7%	30.3 ± 2.7%
Contralateral DRGs	0%	0%	79.9 ± 7.6%	0%	20.1 ± 7.9%

**Table 10 cells-14-00516-t010:** Percentages of the intraganglionic distribution pattern of FB^+^/PNX^+^ neuronal population containing CRT and located in the ipsi- and contralateral DRG studied. The data obtained from all studied animals were pooled and presented as mean ± SD, with *N* = 6 animals.

DRG Subdomain	P	Cr	Cd	Cn	Md
Ipsilateral DRGs	10.5 ± 9.1%	0%	54.7 ± 10.8%	5.2 ± 4.5%	29.6 ± 4.5%
Contralateral DRGs	0%	0%	96.3 ± 3.7%	0%	3.7 ± 3.7%

**Table 11 cells-14-00516-t011:** Percentages of intraganglionic distribution pattern of FB^+^/PNX^+^ neuronal populations containing SOM and located in the ipsilateral and the contralateral DRG studied. The data obtained from all studied animals were pooled and presented as mean ± SD, with *N* = 6 animals.

DRG Subdomain	P	Cr	Cd	Cn	Md
Ipsilateral DRGs	11.5 ± 5.1%	32.8 ± 4.0%	55.7 ± 8.0%	0%	0%
Contralateral DRGs	4.1 ± 5.5%	0%	95.9 ± 5.5%	0%	0%

**Table 12 cells-14-00516-t012:** The relative numbers of individual, differently neurochemically coded, subpopulations of PNX^+^ sensory neurons supplying the urinary bladder, as determined by the analysis of consecutive serial sections of the studied ganglia.

Collocation Patterns of PNX with Different Neurotransmitters in the Bladder DRG Neurons	%
PNX^+^/SP^+^	14.6 ± 0.9
PNX^+^/GAL^+^	3.7 ± 0.8
PNX^+^/CGRP^+^	4.2 ± 0.8
PNX^+^/SP^+^/PACAP^+^	3.5 ± 1.1
PNX^+^/SP^+^/GAL^+^	6.1 ± 1.2
PNX^+^/SP^+^/CGRP^+^	4.3 ± 0.8
PNX^+^/SP^+^/CRT^+^	0.7 ± 0.5
PNX^+^/SP^+^/NOS^+^	0.8 ± 0.9
PNX^+^/GAL^+^/CGRP^+^	7.2 ± 0.8
PNX^+^/NOS^+^/PACAP^+^	1.9 ± 1.1
PNX^+^/GAL^+^/PACAP^+^	3.5 ± 1.9
PNX^+^/CRT^+^/CGRP^+^	3.7 ± 1.3
PNX^+^/PACAP^+^/CGRP^+^	2.2 ± 1.4
PNX^+^/SP^+^/CGRP^+^/PACAP^+^	6.7 ± 2.1
PNX^+^/SP^+^/GAL^+^/PACAP^+^	3.2 ± 1.5
PNX^+^/SP^+^/GAL^+^/CGRP^+^	4.4 ± 0.6
PNX^+^/SP^+^/GAL^+^/CRT^+^	8.1 ± 1.8
PNX^+^/SP^+^/CGRP^+^/CRT^+^	0.7 ± 0.2
PNX^+^/SP^+^/NOS^+^/PACAP^+^	2.2 ± 0.8
PNX^+^/SP^+^/SOM^+^/CGRP^+^	0.7 ± 0.7
PNX^+^/NOS^+^/PACAP^+^/CGRP^+^	3.0 ± 1.3
PNX^+^/SP^+^/PACAP^+^/CRT^+^/CGRP^+^	2.2 ± 0.9
PNX^+^/SP^+^/GAL^+^/PACAP^+^/CGRP^+^	8.5 ± 2.8
PNX^+^/SP^+^/NOS^+^/PACAP^+^/CGRP^+^	1.2 ± 0.7
PNX^+^/SP^+^/GAL^+^/NOS^+^/CGRP^+^	0.6 ± 0.3
PNX^+^/SP^+^/SOM^+^/GAL^+^/PACAP^+^/CGRP^+^	2.1 ± 0.8

## Data Availability

The data that support the findings of this study are available from the corresponding author upon reasonable request.

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
