# Peer review of "Distribution and Neurochemical Characterization of Dorsal Root Ganglia (DRG) Neurons Containing Phoenixin (PNX) and Supplying the Porcine Urinary Bladder"

_cells, 2025, doi:10.3390/cells14070516_

Round 1
Reviewer 1 Report
Comments and Suggestions for Authors
I read with interest this manuscript. The study idea is good and the methodology is well described. The conclusions are in line with the study aims. I have few minor comments:
- please remove the following sentences from the discussion "Authors should discuss the results and how they can be interpreted from the perspective of previous studies and of the working hypotheses. The findings and their implications should be discussed in the broadest context possible. Future research directions may also be highlighted".
- please consider sub-sections of the discussion in order to make easier to read the paper
- please consider a sub-section regarding "future perspective and clinical implicatiosn"
Good.
Reviewer 2 Report
Comments and Suggestions for Authors
All concerns:
- Firstly, these authors have to verify the feature of nerve fibers between the bladder and those DRGs; then based upon the afferents raised from the bladder or the efferents from those DRGs to the bladder to decide the term of prograde or retrograde accordingly;
- please use the same term for DRG or sensory neurons, rather than nerve cells or sensory cells;
- generally, so called sensory means afferents, the term of projection/projecting means efferents; so please tell exactly the feature (afferents of efferents) of those fiber projecting to the bladder; otherwise, these authors mentioned DRGs should receive an afferent inputs raised from the bladder, rather that projecting to the bladder;
- line-74: the porcine bladder receives dual afferents is a conceptual error; in contrast, the sensory information is initiated from the bladder (one of an internal organs) and send these afferent inputs into those DRGs as mentioned locations;
- line-129: due to the mentioned conceptual error, these author thought the inflorescent tracer is retrogradely transported to the DRGs, actually here should be prograde because of their afferent feature; tracer itself could be termed as prograde or retrograde based on the feature of fibers or axonal flow direction (down or against);
- line-130: bladder projecting neurons should be modified accordingly;
- based upon the conduction velocity of afferent fibers, they are generally classified as myelinated and unmyelinated afferents, which is an unique way to verify the feature of sensory fibers, the size of neurons sometime overlapped among the classification; due to the technical reason and the distance between DRGs and related internal organs, it is no way to quantify the conduction, however, some other way could be used for the fiber classification, for example to use HCN1 antibody to label the myelinated fibers or IB4 for unmyelinated ones with functional identification, which are for sure to be better than the size;
- partially data presented in the current observation have been published by the same corresponding author, exactly the same for the figure 2;
- most importantly, this is descriptive manuscript with only one dimension, so their data have very limited functional interpretation of those DRGs.
- the first paragraph is kind strange, it seems the reviewer/editors comment or suggestion, make no sense to the discussion;
- internal organs also send their visceral afferent information to the nodose ganglia through Vagus, just want to know if the fluorescent signal could be detected in the nodose neurons, the majority fibers within the Vagus are sensory and it would be easy to determine the conduction velocity due to the enough distance from the bladder to the nodose ganglia and this would grant the functional explanation of their finding;
- with their long discussion, the large parts of it are not based on their current investigation.
Round 2
Reviewer 2 Report
Comments and Suggestions for Authors
No further comments.